# Smart Indicator Film Based on Sodium Alginate/Polyvinyl Alcohol/TiO_2_ Containing Purple Garlic Peel Extract for Visual Monitoring of Beef Freshness

**DOI:** 10.3390/polym15214308

**Published:** 2023-11-02

**Authors:** Kai Jiang, Jiang Li, Margaret Brennan, Charles Brennan, Haiyan Chen, Yuyue Qin, Mingwei Yuan

**Affiliations:** 1Faculty of Food Science and Engineering, Kunming University of Science and Technology, Kunming 650550, China; jjking2101@163.com (K.J.); lj18855035416@163.com (J.L.); seacome@163.com (H.C.); 2School of Science, Royal Melbourne Institute of Technology University, Melbourne 3000, Australia; margaret.brennan@lincoln.ac.nz (M.B.); charles.brennan@rmit.edu.au (C.B.); 3Green Preparation Technology of Biobased Materials National & Local Joint Engineering Research Center, Yunnan Minzu University, Kunming 650500, China

**Keywords:** smart packaging, anthocyanins, color response, beef freshness, indicator film

## Abstract

The aim of this study was to prepare a novel pH-sensitive smart film based on the addition of purple garlic peel extract (PGE) and TiO_2_ nanoparticles in a sodium alginate (SA)/polyvinyl alcohol (PVA) matrix to monitor the freshness of beef. FT-IR spectroscopy revealed the formation of stronger interaction forces between PVA/SA, PGE, and TiO_2_ nanoparticles, which showed good compatibility. In addition, the addition of PGE improved the tensile strength and elongation at break of the composite film, especially in different pH environments, and the color response was obvious. The addition of 1% TiO_2_ nanoparticles significantly improved the mechanical properties of the film, as well as the light barrier properties of the film. PGE could effectively be uniformly dispersed into the composite film, but it also had a certain slow-release effect on the release of PGE. PGE had high sensitivity under different pH conditions with rich color changes, and the color showed a clear color change from red to yellow-green when the pH increased from 1 to 14. The same change was observed when it was added to the film. In particular, by applying this film to the process of beef preservation, we judged the freshness of beef by monitoring the changes in the TVB-N value and pH value during the storage process of beef and found that the film showed obvious color changes during the storage process of beef, from blue (indicating freshness) to red (indicating non-freshness), and finally to yellow-green (indicating deterioration), which indicated that the color change of the film and the freshness of the beef maintained a highly consistent.

## 1. Introduction

An essential source of protein, fat, and other nutrients for human survival is fresh meat [1]. Fresh meat, however, is extremely vulnerable to deterioration because of microbial contamination, protein oxidation, and lipid oxidation [2]. In particular, improper transportation and storage methods by consumers can easily lead to foodborne illness. And some fresh meat that is still safe to eat but nearing its shelf life is easily discarded, resulting in food waste [3]. In addition, traditional methods for testing the physical and chemical properties, microbiological characteristics, and freshness of meat are difficult and cumbersome for the average consumer and seller to perform in real time, which can further lead to food safety issues and meat waste problems [4]. Therefore, it is important to develop a product that can monitor the shelf life and quality of fresh meat. In the case of fresh meat, the pH of the storage environment changes during storage due to the oxidative degradation of proteins and fats caused by the action of microorganisms and endogenous enzymes, which increase the content of total volatile basic nitrogen (TVB-N). This makes it possible for pH colorimetric indicators to play a huge role in the field of smart indicator packaging materials for the real-time monitoring of meat freshness [3,5].

Anthocyanins are water-soluble colorants widely found in the petals and peels of colored plants and have good pH-responsive properties [6]. The different color changes of anthocyanins in aqueous solutions depend mainly on the structural changes of anthocyanins and are particularly susceptible to acidic and alkaline environments [7,8,9]. Unlike synthetic pigments with potential toxicity risks (bromophenol blue, methyl red, and polyaniline), non-toxic, safe natural dyes have been used as alternatives for indicator packaging materials [10,11]. Researchers have already prepared pH-indicating films by loading anthocyanins into polymeric matrices. For example, Li et al. [12] found that the addition of blue corn anthocyanin to corn starch/polyvinyl alcohol-based films was useful for tilapia preservation applications. Kim et al. [13] monitored shrimp freshness by adding *Clitoria ternatea* flower anthocyanin and zinc oxide nanoparticles to a gelatin/agar matrix. Garlic peel, as a by-product of garlic, may be a bio-resource with recycling potential. Some studies have found that garlic peel contains anthocyanins that have antioxidant activity and antibacterial properties [14]. It has also been used to prepare porous carbon for CO_2_ adsorption [15]. Purple garlic peel is more colorful, and purple garlic extract (PGE) is a promising colorant and antioxidant, and no application of purple garlic extract in edible polysaccharide composite films has been reported.

Research on food packaging has recently focused on smart indicator packaging made of natural materials. Natural polymer materials, such as xanthan gum, sodium alginate, pectin, polyvinyl alcohol, and chitosan, are among them. They are readily available, biodegradable, and satisfy the requirements of customers for goods that are safe [14]. Sodium alginate (SA) is highly degradable and has good compatibility, but it cannot be widely used in films due to its low stability [16]. Polyvinyl alcohol (PVA), on the other hand, has better film-forming properties and strong adhesive properties, which can be combined with sodium alginate to improve film properties [17]. And TiO_2_ nanoparticles have been widely used in food packaging due to their better stability, compatibility [18,19], UV-blocking properties [20], and certain loading functions [21].

In this study, the properties of the films were improved by adding PGE as a color indicator and TiO_2_ to the SA/PVA film matrix. The color development properties of the PGE anthocyanin color indicator were investigated, and the properties of the indicator film were characterized, as well as the color development response of the indicator film. The PGE indicator film was also applied to fresh beef to validate its usefulness for beef freshness indicators.

## 2. Materials and Methods

### 2.1. Materials

Purple-skinned garlic and fresh beef were purchased from Kunming Public Vegetable Market (Kunming, China). Sodium alginate (SA), polyvinyl alcohol (PVA) with a viscosity of 54.0–66.0 mPa·s, and TiO_2_ nanoparticles (>99.8%) were obtained from Aladdin Biochemical Science and Technology Corporation (Shanghai, China). Hydrochloric acid (HCl) and sodium hydroxide (NaOH) were purchased from Sinopharm Chemical Reagent Co. (Shanghai, China). All other reagents were analytical grade.

### 2.2. Preparation of Purslane Garlic Extract (PGE)

First, fresh Purslane garlic was manually peeled after washing and dried under natural conditions. Then, the dried garlic skin was pulverized with an SMF2004 mill to obtain garlic skin powder. Minor modifications were made based on the original methodology [22]. Garlic peel powder was added to 70% ethanol solution (1/10 *w*/*w*) and left overnight. The solution was ultrasonicated for 30 min, and then the solid waste was filtered out. The filtrate was rotary evaporated using a rotary evaporator (LC-N-1100D, Shanghai Li-Chen Instrumentation Science and Technology Co., Ltd., Shanghai, China) at 40 °C for 2 h to concentrate the extract. The above extract was freeze-dried in a lyophilizer (YTLG-20FDT, Shanghai Yetuo Technology Co., Ltd., Shanghai, China) to obtain the PGE powder.

### 2.3. UV-Vis Spectral Analysis of PGE Solutions

The methodology was carried out according to Luchese et al. [23]. The above extracts were adjusted to the desired pH of the PGE solution by adding 0.1 M HCl and NaOH, and the color change of the PGE solution in different pH ranges was recorded. And the UV-visible spectra of PGE solutions with pH were measured in the range of 400–700 nm using a UV-visible photometer (Shimadzu, UV-1800, Kyoto, Japan). pH range (1–14) to ensure a broad spectrum of color changes [22,24].

### 2.4. Preparation of Composite Smart Indicator Packaging Film

These films were based on the original method and continue to be improved upon [25]. A total of 1.75 g SA powder was added to 70 mL of distilled water and stirred thoroughly for 90 min to make a homogeneous mixture. Meanwhile, 0.75 g of PVA was added to 30 mL of distilled water and stirred thoroughly at 90 °C in a magnetic stirring water bath. After cooling to room temperature, the two solutions were mixed (SA/PVA 7/3 *w*/*w*) and stirred for 30 min. And 20 wt% glycerol was added as a plasticizer and mixed thoroughly again. Different concentrations (10 and 15 wt%) of PEG powder and TiO_2_ nanoparticles (0.5 and 1 wt%) were added to the mixture, and all mixtures were sonicated for 30 min to obtain close to 100 mL of film-forming solution. Finally, the film-forming solution was poured into a Plexiglas plate (20 cm × 20 cm) and dried for 12 h at 35 °C and 40% relative humidity. The composite films containing 0%, 10%, and 15% PGE were denoted as SP, SPG10, and SPG15, respectively. The PGE composite films containing 0.5% TiO_2_, and 1% TiO_2_ were named SPG10T_0.5_, SPG10T_1_, SPG15T_0.5_, and SPG15T_1_, respectively.

### 2.5. Performance Testing of Films

#### 2.5.1. Thicknesses

The thickness of the film samples was measured using a digital micrometer (DL9325, Shanghai Shen Han Gauge Co., Ltd., Shanghai, China) with an accuracy of 0.001 mm. Measurements were taken at five different random locations, and averages were calculated.

#### 2.5.2. Moisture Content

The 20 mm × 20 mm film samples were equilibrated in a desiccator at room temperature for 48 h and then dried to constant weight at 105 °C. The formula for calculating the moisture content (*Mc*) was as follows:(1)$$Mc\;\left(\%\right)=\frac{M_{i}-M_{f}}{M_{i}}\times 100$$
where *M_i_* is the initial weight (g) of the film, and *M_f_* is the weight (g) after drying.

#### 2.5.3. Mechanical Properties

According to the Chinese national standard (GB/T 1040.3-2006 [26]), the mechanical properties of the films were determined by tensile testing with an electronic universal testing machine (SANS CMT 4104, SANS Enterprise Development Co., Ltd., Shanghai, China). The sample films were cut into long strips of 20 mm × 100 mm, fixed on fixtures 30 mm apart, and then tested at 5 mm/s for the experiment. Tensile strength (TS) and elongation at break (EB) were obtained directly from the software.

#### 2.5.4. Scanning Electron Microscope (SEM)

The composite film was cut into 10 mm × 10 mm squares and frozen in liquid nitrogen. Cross-sections of the film samples were bonded to a metal sample stage using a double-sided conductive adhesive and sprayed with gold. Micrographs of cross-sections taken at random locations were observed using a field emission scanning electron microscope (Nova Nano SEM 450) at an accelerating voltage of 10 kV.

#### 2.5.5. Fourier Transform Infrared Spectrogram (FTIR)

The films were characterized using a Nicolet 5700 FTIR spectrometer. The samples were sheared to a size of 20 mm × 20 mm, wiped with alcohol, and tested. The resolution was 4 cm^−1^, and the spectra were recorded in the range of 400–4000 cm^−1^.

#### 2.5.6. Color Difference in Film

The color of the film was measured using a WSC-S colorimeter, which expresses the color of the film as three parameters, *L**, *a**, and *b**, and calculates the total color change Δ*E** value according to the following formula:(2)$$∆E*=\sqrt{{\left(L_{0}-L*\right)}^{2}+{{(a}_{0}-a*)}^{2}+({b_{0}-b*)}^{2}}$$
where *L** (91.29), *a** (−0.89), and *b** (3.60) are whiteboard values used for calibration, and *L*_0_, *a*_0_, and *b*_0_ are measured values of the film samples.

#### 2.5.7. UV Transmittance

A UV-visible spectrophotometer was used to measure the UV and visible resistive transmittance of the films at wavelengths of 200–800 nm.
(3)$$Tr=\frac{log\%T600}{d}$$
where *Tr* denotes transparency, %*T*600 denotes the transmittance of the film at 600 nm, and *d* denotes the film thickness.

#### 2.5.8. Migration Experiments with PGE

A 50% ethanol solution was chosen as a low-fat food analog for the migration of PGE in low-fat foods in films [11,27]. PGE solutions of different concentrations (0.1, 0.2, 0.3, 0.4, 0.5 mg/mL) were first prepared, and the absorbance of the solutions was recorded at 520 nm using a spectrophotometer to produce a standard curve. The films (2.5 mm × 2.5 mm) were immersed in 2 mL of 50% ethanol solution in a room temperature environment (25 ± 2 °C), and the absorbance at 520 nm was recorded as a function of time (0, 30, 60, 90, 120 min). The release rate of PGE was obtained from the standard curve [28].

#### 2.5.9. Analysis of the Color Development Response of Films at Different pHs

The films (SP, SPG10T_1_, SPG15T_1_) were cut into 4 cm × 4 cm and immersed in buffer solution (pH = 1–14) for 5 min. Color changes were recorded with a digital camera.

### 2.6. Film Application Tests on Beef

Fresh beef was cut into uniformly sized and shaped pieces (20 g) using an aseptic knife and placed in sterilized PET plastic boxes (140 mm × 100 mm × 50 mm), with each box containing 4 pieces of beef. The box was sealed with SPG10T_1_ film and stored in a refrigerator at 4 ± 1 °C. Data were collected from the samples at 0 d, 2 d, 4 d, and 6 d.

#### 2.6.1. Determination of pH in Beef

According to the Chinese national standard (GB 5009.237-2016 [29], the Ph level was tested. The samples were churned using a high-speed rotary cutter, followed by measurement of the meat paste using a Ph meter (METTLER TOLEDO FE-28) [30]. And the collection of photographs was synchronized to indicate the label card preservation experiment.

#### 2.6.2. Determination of Volatile Salt Nitrogen Content in Beef (TVB-N)

TVB-N content was determined according to the semi-trace nitrogen determination method of the Chinese national standard (GB 5009.228-2016) [31,32]. The sample beef was stirred well and weighed accurately to 20 g, accurate to 0.001 g. A volume of 100 mL of trichloroacetic acid solution was added, shaken to mix well, macerated for 30 min, and then filtered. Then, 10 mL of filtrate, 10 mL of H_2_BO_2_ (20 g/L), and 5 mL of MgO (10 g/L) were accurately pipetted together and added to the reaction chamber to start distillation. The indicator was a mixture of 1 part ethanol solution of methyl red and 5 parts ethanol solution of bromocresol green. The distillation was terminated by titration using 0.01 mol/L HCL, while distilled water was used as a blank control. The formula for calculating the volatile saline nitrogen content was as follows [33]:(4)$$X=\frac{\left(V_{1}-V_{2}\right)\times c\times 14}{m\times \left(V/V_{0}\right)}\times 100$$
where *X* is the TVB-N of the sample, mg/100 g; *V*_1_ is the volume of HCL consumed by the sample, mL; *V*_2_ is the volume of HCL consumed by the blank control, mL; *V* is the filtrate volume, mL; *V*_0_ is the total volume of the sample solution in milliliters (mL); *c* is the concentration of the HCL solution, mol/L; and *m* is the sample mass, g.

### 2.7. Statistical Analysis

To guarantee the correctness of the results, three duplicates of each sample were obtained. SPSS 10.0 was used for the statistical studies, which included Duncan’s multiple range testing (*p* < 0.05) and an analysis of variance (ANOVA). Data were expressed as the mean ± standard deviation. Additionally, the data were visualized using Origin 9.0. PowerPoint 2020 was used to plot the graphical abstract.

## 3. Results and Discussion

### 3.1. pH Sensitivity of PGE Solutions

The color change of the PGE solution in different pH ranges is recorded in Figure 1a. The specific color development was as follows: PGE was red, light red, pink, blue cyan, green, and yellow in pH ranges of 1–2, 3–10, 11, 12, 13, and 14, respectively.

The UV-Vis spectra of the PGE solutions at various pH levels are shown in Figure 1b,c. As the pH increased from 1 to 14, the greatest absorption peak of PGE migrated from the region around 510 nm to the region around 615 nm. A bathochromic shift is the name for this change, which often occurs in anthocyanins [22,34]. When the PGE solution was at pH 1–5, the maximum absorption peak occurred at 500–525 nm and decreased with increasing pH. Surprisingly, the maximum absorption peak shifted to around 610 nm when the pH was greater than 6, and instead increased with increasing pH. Such a change may be due to the change in the chemical structure of PGE anthocyanins in different pH environments [35]. In an acidic environment, anthocyanin exists mainly as a yellow molten cation, at which time the PGE solution is red. When the solution shifts from a weak acid, methanol pseudobases (colorless) begin to appear in the PGE anthocyanin structure and the color [36]. Zeng et al. reported similar results for color changes and chemical structure transformations of mulberry anthocyanins in different pH solutions [37]. Finally, the central ring of the anthocyanin is broken down under strong alkaline conditions to form the chalcone structure, and the color of the solution changes to green and yellow [38,39]. Therefore, PGE solutions have great potential in smart packaging due to their highly sensitive responses under different pH conditions.

### 3.2. Properties of Films

#### 3.2.1. Thicknesses

According to Table 1, when the PGE was raised from 0% to 15%, the thickness of the films rose from 63.02 μm to 71.01 μm. There was a significant increase in thickness when 15% PGE was added (*p* < 0.05). This may be due to an increase in solids, resulting in an increase in film thickness [40]. Zhang et al. [19] showed that adding anthocyanin-rich black plum peel extract to chitosan films as an indicator film resulted in an increase in film thickness. However, the addition of trace amounts of TiO_2_ nanoparticles did not lead to significant changes (*p* < 0.05) in film thickness. This may be due to the loading capacity of TiO_2_ nanoparticles [41] and probably due to the low total additions, resulting in insignificant thickness changes (*p* < 0.05).

#### 3.2.2. Moisture Content

The higher the moisture content of the indicator film, the more hydrophilic the film is and the stronger the interaction with water molecules, which can lead to a sluggish or inaccurate color development response of the film [42]. In Table 1, Mc was significantly lower (*p* < 0.05) for films with PGE anthocyanin added relative to SP films. Relative to SP, SPG10 was reduced by 14.42% and SPG15 by 31.80%, indicating that the strong interaction between SA/PVA and PEG limits the interaction between hydroxyl groups and water molecules in SA/PVA [43], leading to reduced hydrophilicity. And the free anthocyanins replace the free water molecules, leading to a significant reduction in the Mc of the film [28].

#### 3.2.3. Mechanical Properties

The tensile strength (TS) and elongation at break (EB) of the composite film films are listed in Table 1. As the PGE content increased from 0% to 15%, TS increased significantly from 23.84 MPa to 34.29 MPa (*p* < 0.05) and EB increased from 15.26% to 28.63% (*p* < 0.05). This may be due to the abundant hydroxyl structure in anthocyanins creating stronger binding with hydrogen bonds in SA/PVA, resulting in stronger intermolecular interaction forces [44,45].

With the incorporation of TiO_2_ nanoparticles, the TS value of the films increased. The highest value of 39.52 Mpa was achieved for SPG15T_1_ film. This may be due to the high surface area and large specific surface area of TiO_2_ nanoparticles, which can promote the interfacial bonding between SA/PVA and TiO_2_ and improve the film strength [18]. Similar findings were made by Liu et al. [46] The addition of TiO_2_ increased the TS value of the PVA/xylan composite film. Meanwhile, the EB values of the films decreased significantly (*p* < 0.05) with the addition of TiO_2_ nanoparticles from 20.54% (SPG10) to 17.29% (SPG10T_1_) and from 28.63% (SPG15) to 24.92% (SPG15T_1_). TiO_2_ increases the rigidity of the film and alters the plasticizing effect of TiO_2_ nanoparticles in the film matrix [19]. Therefore, the appropriate addition of TiO_2_ nanoparticles can improve the mechanical properties of the films.

#### 3.2.4. Microstructure Analysis of Films

The SEM microstructures of the films are shown in Figure 2, and all the films show relatively continuous, crack-free, tightly structured cross-sections. The SP film cross-section exhibits brittle fracture (shown by the red circle in Figure 2a) [17]. However, the brittle fracture disappeared with the addition of PGE, which is consistent with the results obtained for the mechanical properties of the films. With the addition of PGE, the EB increased, and the ductility was stronger. It was shown that free anthocyanins improved the interaction force between SA/PVA, this also contributes to the compatibility between SA/PVA.

With the addition of TiO_2_ nanoparticles, brittle fracture begins to occur (Figure 2d–g), and the cracking increases with the increase in content, which is more pronounced in SPG10T_1_ (Figure 2e) and SPG15T_1_ (Figure 2g). This is similar to the conclusion about the mechanical properties of the films. Surprisingly, the surface of SPG10T_0.5_ (Figure 2h) and SPG15T_0.5_ (Figure 2j) films appeared to have particles with an uneven size distribution, and the distribution of titanium dioxide on the surface of SPG10T_1_ (Figure 2i) and SPG15T_1_ (Figure 2k) was relatively dense. It has been shown that 1% of TiO_2_ nanoparticles can be well distributed in SP films, and the presence of TiO_2_ nanoparticles is also responsible for the increase in the tensile strength and decrease in the light transmittance of the composite films [20]. It can be seen that SA, PVA, PGE, and TiO_2_ can be tightly bound to form a continuous and dense microstructure. These results were also confirmed by the FTIR analysis.

#### 3.2.5. FTIR Spectra of Thin Films

The FTIR spectra of SP, SPG10, SPG15, SPG10T_0.5_, SPG10T_1_, SPG15T_0.5_, and SPG15T_1_ films are shown in Figure 3.

The SP film showed a characteristic peak at 3598 cm^−1^, which is an indication of the presence of hydroxyl groups [47]. The other peaks located at 1592 cm^−1^ can be attributed to the C=C tension vibration, and the characteristic peak at 1037 cm^−1^ is derived from the carbonyl group in sodium alginate [48]. The asymmetric stretching of the C-H bonds in PVA is responsible for the distinctive spectral band at 2954 cm^−1^. The addition of anthocyanin also shifted the corresponding absorption peaks of the composite film. For example, the peak at 1592 cm^−1^ of the SP film shifted to longer wave numbers (1621 cm^−1^, 1643 cm^−1^), suggesting that the incorporation of PGE anthocyanins led to inter- and intramolecular hydrogen-bonding interactions between the film matrices [49]. Due to the presence of TiO_2_ nanoparticles, the intensity of the above band in the range of 1400–1700 cm^−1^ is slightly reduced and shifted, and the characteristic peak at 1643 cm^−1^ of the SPG15 film is shifted to the vicinity of 1475 cm^−1^, which indicates that intermolecular interactions between the functional groups occurred [50]. In contrast to the SP films, no films showed new structural peaks in their spectra, and all showed broad peaks in the 3200–3700 cm^−1^ wave number range due to the stretching vibration of the -OH group [51]. Other characteristic peaks of SP, such as 1153 cm^−1^ (C-O bending vibration), 876 cm^−1^ (C-C stretching vibration), and the incorporation of PGE anthocyanin and TiO_2_ did not change these structures of SP film. The results showed that PGE and TiO_2_ were compatible with SA/PVA and formed films with good mechanical properties through intermolecular interactions.

#### 3.2.6. Color Parameters

The color parameters of the films are shown in Table 2. The addition of PGE resulted in a significant decrease in *L* from 65.32 to 39.28 (*p* < 0.05), *a* decreased from 2.88 to −2.60 (*p* < 0.05), *b* decreased significantly from 3.91 to −2.71 (*p* < 0.05), and Δ*E* increased significantly (*p* < 0.05). The addition of TiO_2_ nanoparticles also caused a change in the color parameters, with a decrease in the *L*, *a*, and *b* values. Asadi and Pirsa found the same result; the incorporation of TiO_2_ nanoparticles decreased the *L*, *a*, and *b* values of the PLA-based films [52]. They attributed this to the effective absorption of UV waves and photocatalytic activity by TiO_2_ leading to the reduction in *a*. The above results show that the films with PGE and TiO_2_ are darkened and more colorful, which is consistent with the appearance of the films.

#### 3.2.7. UV Transmission Rate

UV-visible light has tremendous negative effects on food [53]. It may lead to accelerated oxidative degradation of food and result in loss of nutrients, changes in color, and even the production of toxic and harmful substances. The UV-Vis spectra of the films reflect the light barrier properties of the films. The absorption spectrum is shown in Figure 4.

SP films show high transmittance (70.8–84.8%) in the visible range (400–800 nm). The light transmission of the films decreased significantly (*p* < 0.05) with increasing PGE content. This suggests that the addition of PGE has a light-blocking effect. Phenolic compounds have been reported to have good UV-absorbing ability, and anthocyanins belong to the polyphenol group [54,55].

TiO_2_ with a crystal structure having a metallic nature may lead to a decrease in light transmittance due to the diffraction and reflection of light. In fact, TiO_2_ nanoparticles have a strong diffuse reflection of light at the film interface due to their large specific surface area and strong reflection of light [46]. Therefore, the increase of TiO_2_ nanoparticles decreases the UV transmittance of the films. Both SPG10T_1_ and SPG15T_1_ films have better light-blocking properties, with TiO_2_ nanoparticles blocking more UV rays and thus inducing photocatalysis [56]. He et al. [46] concluded that the addition of TiO_2_ can effectively block light in the UV region. In addition, Fonseca et al. [57] confirmed that gelatin-titanium dioxide films absorbed more light at UV wavelengths ranging from 222 to 380 nm, which involves both UV-A and UV-B types.

#### 3.2.8. PGE Release Rate in Films

The anthocyanin release characteristics of the composite film when applied to beef packaging were evaluated by migration experiments of anthocyanins in a 50% ethanol solution (simulating a low-fat food) (Figure 5).

Figure 5a represents the standard curve for PGE anthocyanins, where the linear regression equation is as follows:(5)$$y=0.1872x+0.0444\;{\;R}^{2}=0.9932$$
where *x* denotes the PGE concentration (mg/mL) *y* denotes the absorbance value.

From Figure 5b, it can be seen that in the films without TiO_2_ nanoparticles incorporated, the release rate was faster in the early stage, the rate slowed down by 30 min, and the release rate was low, only around 60%, where it increased slightly with the increase in PGE. Surprisingly, the release rate decreased with the increase in TiO_2_ nanoparticles, and the release rate reached 70–80%. In particular, the best PGE release effect in SPG10T_1_ and SPG15T_1_ films was around 80%. This may be due to the presence of TiO_2_ nanoparticles that can play a supporting role for the film, which not only can make the PGE better mixed in the film, so that the release rate is higher, but there is also a certain delay in the release of PGE to better improve the effect of the film [21,58].

#### 3.2.9. Analysis of the Color Development Response of Films at Different pHs

From the above, it can be seen that by adding PGE and TiO_2_ nanoparticles to SA/PVA, the film properties were effectively improved, and by the release rate of PGE, we chose SPG10T_1_ and SPG15T_1_ films for the color reaction under different pH conditions, as shown in Figure 6.

The SP film without anthocyanin showed no significant color change in the different pH ranges. In contrast, the color of the structural transformation of anthocyanin due to different acid–base environments subsequently changed, and the extent of the color change varied with the amount of PGE added to the SP composite film [59]. Figure 6 shows that there was a difference in the color change between SPG10T_1_ and SPG15T_1_ films, but it was not significant. Therefore, SPG10T_1_ film was selected as suitable for smart food packaging material.

### 3.3. Film for Freshness in Beef

During the storage of meat, the oxidative decomposition of nitrogen-containing substances (proteins, etc.) produces a large amount of TVB-N, which causes the pH in the packaging environment to change [60]. Changes in TVB-N and pH occur during the storage of beef using SPG10T_1_ as a smart packaging film, as shown in Figure 7. In Figure 7b, the “fresh” TVB-N value in the reference color card is <15 mg/100 g (GB 2707-2016 [61]), “S-Frech” denotes the maximum permissible upper limit, where the TVB-N value is <20 mg/100 g [62], and “spoilage” is the pre-experimentation to make the beef completely spoiled using the color card obtained from the smart indicator film.

The Chinese standard GB 2707-2016 sets the TVB-N limit for fresh meat at 15 mg/100 g. From Figure 7a, it can be seen that both TVB-N and the pH of beef increased significantly with increasing days of storage. For fresh beef (0 day), the values of TVB-N and pH were 8.32 mg/100 g and 5.91, respectively. By day 2, it increased to 12.13 mg/100 g and 6.33, respectively, indicating that the beef was still fresh. And on the 4th day, the TVB-N value of beef reached 19.24 mg/100 g, which exceeded the standard limit and began to deteriorate, but not fresh beef. This corresponds to “S-Fresh” on the color card and is not recommended for consumption. And on the 6th day, the TVB-N value was as high as 24.33 mg/100 g, which far exceeded the standard limit and was completely spoiled and inedible. Surprisingly, the effect of the SPG10T_1_ film can be seen in Figure 7b. The change in color of the film visible to the naked eye during the preservation process further illustrates that SPG10T_1_ film can accurately evaluate the freshness of beef by its sharp color change. We produced a comparative effect graph of spoilage.

### 3.4. Correlation Analysis

The correlation analysis between the properties and color of the composite film is shown in Figure 8.

Surprisingly, there was a negative correlation between Mc and thickness (r = −0.88, *p* < 0.01) and EB (r = −0.99, *p* < 0.001). This may be due to the addition of PGE, which increases the solid mass and the film thickness, as well as with the stronger interaction force between SA/PVA and PGE; EB thus increases, making the force between SA/PVA and water molecules lower and resulting in a decrease in moisture content. Thickness was negatively correlated with *L* (r = −0.96, *p* < 0.001) and *b* (r = −0.97, *p* < 0.001); TS was negatively correlated with *L* (r = −0.95, *p* < 0.01) and *a* (r = −0.97, *p* < 0.001). It was shown that the addition of PGE and TiO_2_ changed the properties of the film while making it darker and more colorful.

## 4. Conclusions

In this study, a novel pH-indicating film was prepared by incorporating PGE and TiO_2_ nanoparticles into SP/PVA matrix and applied to monitor the freshness of beef. Scanning electron microscopy (SEM) analysis showed that 1% TiO_2_ nanoparticles and PGE were uniformly dispersed and formed a dense structure in the SP/PVA matrix. FT-IR spectroscopy showed that SA/PVA, PGE and TiO_2_ nanoparticles interacted with each other to form a stronger force, which demonstrated good compatibility. In addition, the addition of PGE increased the thickness, tensile strength, and elongation at break of the laminated film and decreased the UV-visible transmittance and elongation at break. And the addition of 1% nano-TiO_2_ can significantly improve the mechanical properties of the film, as well as the film’s light barrier properties. The PGE can be more uniformly dispersed into the film and also play a certain role in slow-release control. Experiments showed that the color of the PGE solution showed a clear color change from red to yellow-green when the pH increased from 1 to 14, and a similar color change was observed when the film was made. The pH-sensitive smart visual labels matched beef spoilage time points to a high degree when SPG10T_1_ films were applied to the beef. The film exhibited a distinct color change from blue (indicating freshness) to red (indicating sub-freshness) and finally to yellow-green (indicating spoilage). The SPG10T_1_ film is therefore a natural, sustainable, and biodegradable packaging material with great potential for monitoring the freshness of beef and other finished meat products.

## Figures and Tables

**Figure 1 polymers-15-04308-f001:**
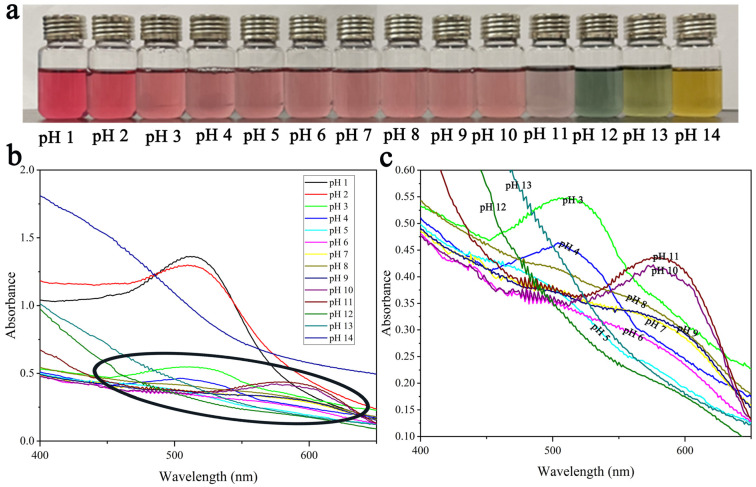
Color change of PGE solutions in the 1–14 pH range (**a**); UV-vis spectra of PGE solutions at different pH conditions (**b**); The black ellipse in (**b**) is enlarged in (**c**).

**Figure 2 polymers-15-04308-f002:**
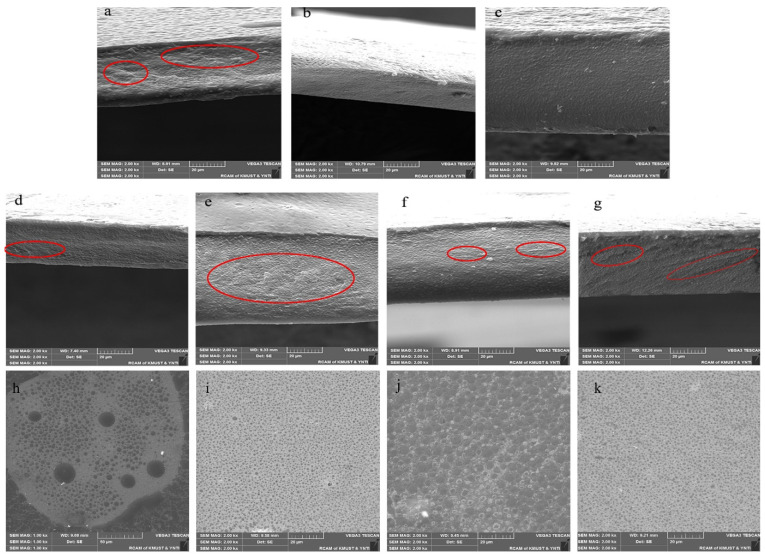
SEM diagram of SP film cross-section (**a**), SPG10 film cross-section (**b**), SPG15 film cross-section (**c**), SPG10T_0.5_ film cross-section (**d**), SPG10T_1_ film cross-section (**e**), SPG15T_0.5_ film cross-section (**f**), SPG15T_1_ film cross-section (**g**), SPG10T_0.5_ film surface (**h**), SPG15T_0.5_ film surface (**i**), SPG15T_0.5_ film surface (**j**), and SPG15T_0.5_ film surface (**k**). Note: Red circles in (**a**,**d**–**g**), indicate significant differences in films.

**Figure 3 polymers-15-04308-f003:**
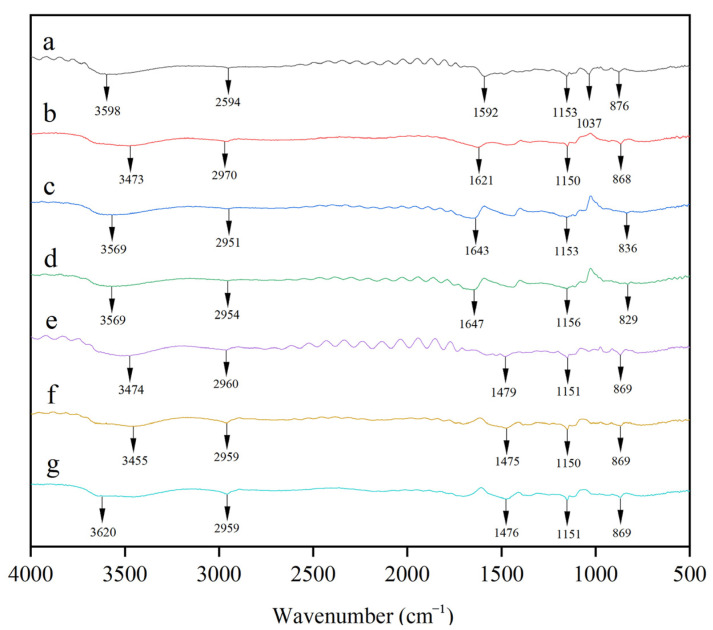
Fourier infrared spectra of SP film (a), SPG10 film (b), SPG15 film (c), SPG10T_0.5_ film (d), SPG10T_1_ film (e), SPG15T_0.5_ film (f), and SPG15T_1_ film (g).

**Figure 4 polymers-15-04308-f004:**
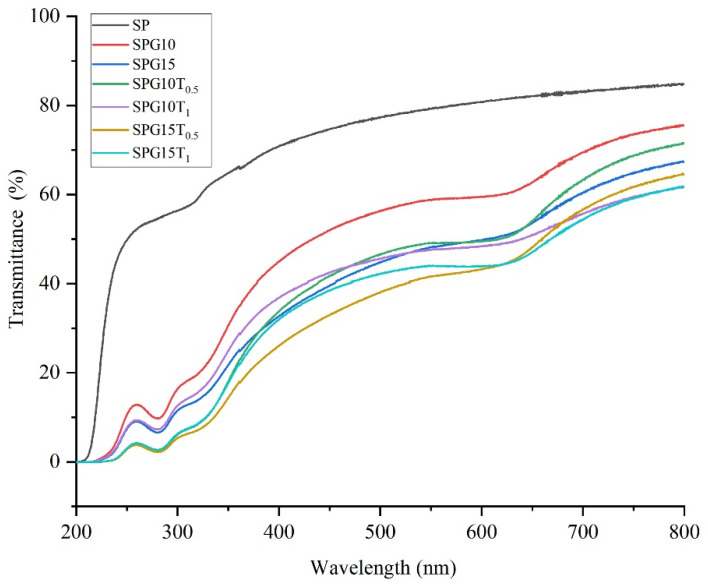
UV-Vis transmission diagram of the composite films.

**Figure 5 polymers-15-04308-f005:**
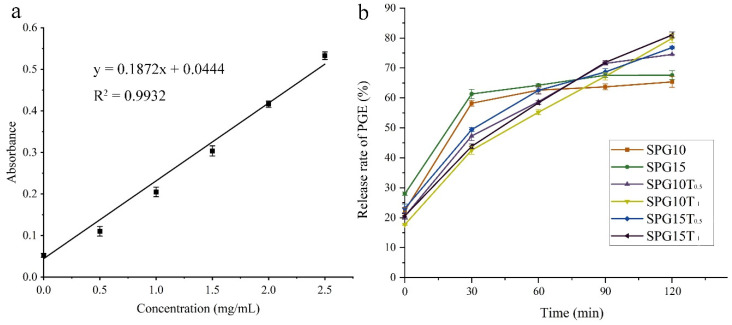
Standard curve of PEG (**a**), release rate of PEG in 50% ethanol solution (**b**).

**Figure 6 polymers-15-04308-f006:**
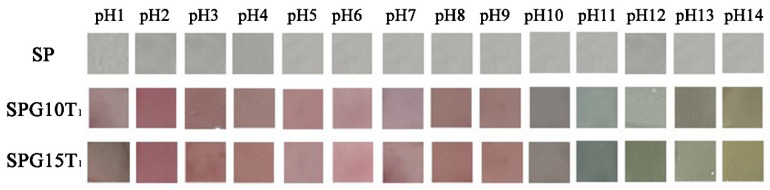
Color change of SPG films in different pH solutions (pH 1–14).

**Figure 7 polymers-15-04308-f007:**
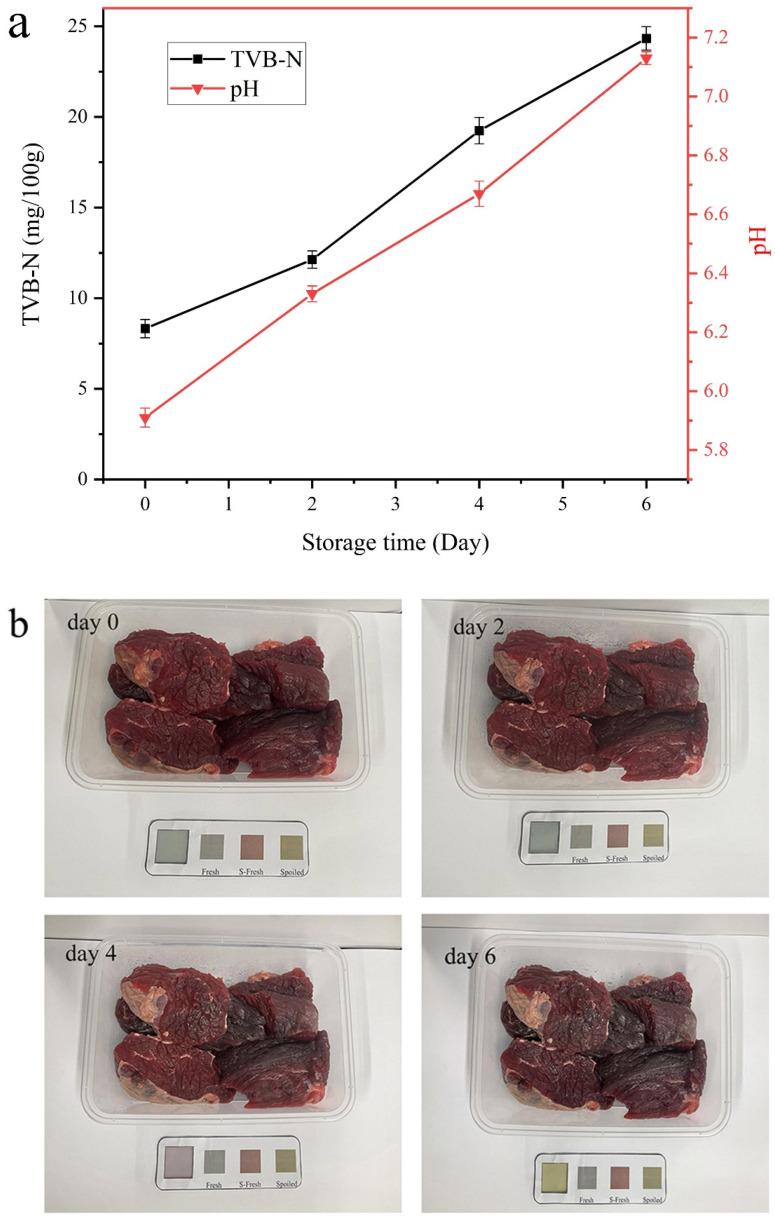
The pH value and TVB-N corresponding to the color change of SPG10T_1_ film during beef storage (**a**) and visual photos of beef (**b**).

**Figure 8 polymers-15-04308-f008:**
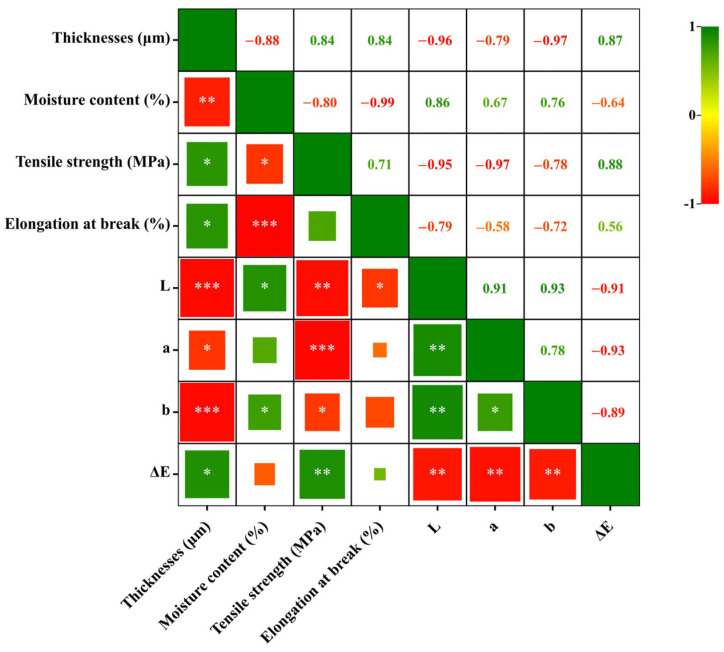
Correlation analysis of composite film properties. NOTE: *, ** and *** indicate significance *p* < 0.05, *p* < 0.01 and *p* < 0.001, respectively.

**Table 1 polymers-15-04308-t001:** Physical properties of composite films.

Films	Thicknesses (μm)	Moisture Content (%)	Tensile Strength (MPa)	Elongation at Break (%)
SP	63.02 ± 3.54 ^a^	19.97 ± 1.08 ^a^	23.84 ± 0.86 ^a^	15.26 ± 1.17 ^a^
SPG10	68.81 ± 4.66 ^ab^	17.09 ± 0.51 ^b^	28.75 ± 0.57 ^b^	20.54 ± 0.74 ^c^
SPG15	71.01 ± 6.28 ^b^	13.62 ± 0.52 ^c^	34.29 ± 0.94 ^d^	28.63 ± 0.48 ^f^
SPG10T_0.5_	68.62 ± 3.51 ^ab^	17.64 ± 0.46 ^b^	31.83 ± 0.46 ^c^	19.56 ± 0.40 ^c^
SPG10T_1_	68.04 ± 4.64 ^ab^	18.20 ± 0.42 ^b^	34.26 ± 0.36 ^d^	17.29 ± 0.86 ^b^
SPG15T_0.5_	70.05 ± 1.58 ^b^	13.73 ± 0.49 ^c^	37.05 ± 0.61 ^e^	27.18 ± 0.49 ^e^
SPG15T_1_	70.43 ± 4.67 ^b^	14.40 ± 0.59 ^c^	39.52 ± 0.60 ^f^	24.92 ± 0.20 ^d^

Different letters indicate significant (*p* < 0.05) differences between two groups. a–f in the same column is marked from small to large.

**Table 2 polymers-15-04308-t002:** Color of sodium alginate/polyvinyl alcohol films containing purple garlic peel extract PGE.

Films	*L*	*a*	*b*	Δ*E*	The Color of Film
SP	65.32 ± 1.18 ^e^	2.88 ± 0.35 ^f^	3.91 ± 0.22 ^c^	26.17 ± 1.17 ^a^	
SPG10	46.49 ± 1.22 ^d^	0.46 ± 0.33 ^e^	−1.50 ± 0.38 ^b^	44.77 ± 1.20 ^b^	
SPG15	39.28 ± 1.89 ^b^	−2.60 ± 0.47 ^cd^	−2.71 ± 0.45 ^a^	51.99 ± 1.89 ^d^	
SPG10T_0.5_	44.88 ± 0.92 ^cd^	−1.89 ± 0.52 ^d^	−1.36 ± 0.26 ^b^	46.39 ± 0.91 ^bc^	
SPG10T_1_	43.92 ± 0.98 ^c^	−3.76 ± 0.23 ^b^	−1.42 ± 0.19 ^a^	54.76 ± 0.72 ^e^	
SPG15T_0.5_	36.55 ± 0.74 ^a^	−3.23 ± 0.37 ^bc^	−2.02 ± 0.56 ^b^	47.37 ± 0.98 ^c^	
SPG15T_1_	34.49 ± 1.09 ^a^	−5.08 ± 0.57 ^a^	−1.76 ± 0.38 ^a^	56.90 ± 1.13 ^f^	

Different letters indicate significant (*p* < 0.05) differences between two groups. a–f in the same column is marked from small to large.

## Data Availability

The data presented in this study are available on request from the corresponding author.

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
