# Peer review of "Smart Indicator Film Based on Sodium Alginate/Polyvinyl Alcohol/TiO2 Containing Purple Garlic Peel Extract for Visual Monitoring of Beef Freshness"

_polymers, 2023, doi:10.3390/polym15214308_

Round 1

Reviewer 1 Report

Comments and Suggestions for Authors

The authors present an interesting SA/PVA composite material enriched with PGE and TiO2 nanoparticles for smart packaging application and I really would like to congratulate with the authors for the precise description of both the methods and procedures employed and the detailed discussion of the results, always improved with comparisons with the existing literature. The presented work is interesting, well-presented and clear and only minor revisions should be done to further improve the manuscript before publication:

ABSTRACT

1) I would not say that thickness is improved, it can be increased or decreased but I see no reasons to say that higher or lower thickness values are better

2) "The pH-sensitive film showed smart visual labeling and beef deterioration time point in line with the higher degree" this sentence is unclear to me...please clarify

3) "The film exhibits a distinct color variation from blue (indicating freshness) to red (indicating sub-freshness) and finally to yellow-green (indicating spoilage)." If we talk about freshness/spoilage it is better to hint here at the independent method you used to discriminate between fresh and spoilt samples. If not specified, the readers may think you are just guessing while this is not the case.

INTRODUCTION

1) Bromocresol blue doesn' exist; you may mean bromocresol green, bromocresol purple, bromophenol blue, bromothymol blue or sulfonephthaleins in general

2) "Kim et al [13] Clitoria ternatea flower anthocyanins were used..." this sentence makes no sense, I think a typo may have occurred

MATERIALS AND METHODS

1) Section 2.3: here you say that the exact pH is reache by adding NaOH or HCl in no buffered solution while in the results you talk about different buffers. Please correct this error

2) Section 2.4: a) what do you mean by 90°C "ambient conditionds"? and b) clearly after stirring for 30 minutes the solution is mixed well, there's no need to specify and, furthermore, it's not very scientifically sounding the sentence "mixed well"

 3) Both in solution and in films, you work in a pH range 1-14 while you have mentioned in the introduction that anthocyanins are stable neither at acidic nor at alkaline pH. Please address this issue

RESULTS

1) Figure 1: why UV-Vis Spectra are cut at 400 nm? since you observe yellow colours, it may be useful to go down to 2350 nm at least

2) A detailed discussion of the spectra in Figure 1 is reported but actually this Figure is very difficult to interpret because all the spectra are confused together apart from those at extreme pHs, that present very high A. Please either split into two figures or use secondary axes to allow the readers to better analyse also the other spectra unless your description cannot be verified looking at the plot.

3) Section 3.2.1: the differences in thickness don't seem to significant looking at the standard deviations (and in fact no p-value is given, differently from the tensile test results). If no significant difference is encountered, please specify this point

4) Table 2 might be enriched adding also the figures of the films

5) Spectra in Figure 4 are acquired on one single sample per type and seem very similar. I don't think you can assume significant differences relying on such similar spectra without any replicate. Please fix this inaccuracy

6) Figure 7: how do you defined the reference colour printed next to the sensor? Why the sub-freshness colour is never observed?

7) Just a comment for future works: section 3.4 can be much more easily and efficiently performed using simple chemometric tools like PCA. Keep it in mind for the future

Comments on the Quality of English Language

Ok

Author Response

Dear reviewers,

Thank you very much for your kindly comments on our manuscript. There is no doubt that these comments are valuable and very helpful for revising and improving our manuscript. In what follows, we would like to answer the questions you mentioned and give detailed account of the changes made to the original manuscript.

Yours sincerely,

Yuyue Qin

Reviewer 2 Report

Comments and Suggestions for Authors

This paper presented on the application of anthocyanin derived from purple garlic extract as a bioindicator in sodium alginate/polyvinyl alcohol/TiO2 film for smart indicator packaging. However, several point should be improved to be published in Polymers.

1. Title should be changed for more understanding. The paper's title must be consistent with your studies' objective and novel findings. This paper examined the utilization of garlic extract as a bioindicator to be supplemented in the SA/PVA film. Moreover, the results also indicated the freshness of meat by using your films. Therefore, please indicate the finding and expected application of the film in the title for insight to all readers.

2. Introduction must add the detail of TiO2. Why did you mix TiO2 into the film? Moreover, the detail of bioindicators in the composition of garlic extract should be added.

3. Methodologies are interesting and clear but some sentences are not completed. Please check grammar throughout your manuscript.

Moreover, could you provide the phycocyanin content or others indicator substance in the garlic extract?

4. Results and Discussion

4.1 Why the line of pH4 (Blue line) showed different pattern from other lines? Please discuss about this event?

4.2 Why TiO2 nanoparticles was added into the film? Do you any reasons for this? Please add.

4.3 The important of FTIR peak should be shown in the table to compare your results. Please add table.

4.4 What is the scientific relationship between the result of slow release of anthocyanin from the film and other results? Please describe.

Author Response

(The authors gave the same response as above.)

Round 2

Reviewer 2 Report

Comments and Suggestions for Authors The authors did the necessary changes in the manuscript. I suggest the acceptance of the manuscript.